# The Role of Fluorine in F-La/TiO_2_ Photocatalysts on Photocatalytic Decomposition of Methanol-Water Solution

**DOI:** 10.3390/ma12182867

**Published:** 2019-09-05

**Authors:** Miroslava Edelmannová, Lada Dubnová, Martin Reli, Vendula Meinhardová, Pengwei Huo, Urška Lavrenčič Štangar, Libor Čapek, Kamila Kočí

**Affiliations:** 1Institute of Environmental Technology, VŠB-Technical University of Ostrava, 17. listopadu 15/2172, 70800 Ostrava, Czech Republic (M.E.) (M.R.); 2Faculty of Materials Science and Technology, VŠB-Technical University of Ostrava, 17. listopadu 15/2172, 70800 Ostrava, Czech Republic; 3Faculty of Chemical Technology, University of Pardubice, Studentská 573, 53210 Pardubice, Czech Republic (L.D.) (V.M.) (Č.C.); 4School of Chemistry and Chemical Engineering, Jiangsu University, 301 Xuefu Road, Zhenjiang 212013, China; 5Faculty of Chemistry and Chemical Technology, University of Ljubljana, Večna pot 113, P.O Box 537, 1001 Ljubljana, Slovenia

**Keywords:** hydrogen production, fluorine, lanthanum, titanium dioxide

## Abstract

F-La/TiO_2_ photocatalysts were studied in photocatalytic decomposition water-methanol solution. The structural, textural, optical, and electronic properties of F-La/TiO_2_ photocatalysts were studied by combination of X-ray powder diffraction (XRD), nitrogen physisorption, Ultraviolet–visible diffuse reflectance spectroscopy (UV-Vis DRS), Electrochemical impedance spectroscopy (EIS), and X-ray fluorescence (XPS). The production of hydrogen in the presence of 2.8F-La/TiO_2_ was nearly up to 3 times higher than in the presence of pure TiO_2_. The photocatalytic performance of F-La/TiO_2_ increased with increasing photocurrent response and conductivity originating from the higher amount of fluorine presented in the lattice of TiO_2_.

## 1. Introduction

Lately, hydrogen has acquired significant attention as a next-generation energy carrier. Photocatalytic water splitting into hydrogen and oxygen in the presence of semiconductor photocatalysts is an attractive process of how to produce it. There are many reviews focusing on this topic every year, e.g., [1,2,3]. Higher efficiency of hydrogen production has been achieved in photocatalytic water splitting under presence of sacrificial agent, among others, preventing the recombination of electron-hole pairs [1].

Since the discovery of water splitting on TiO_2_ electrodes in the presence of UV irradiation, TiO_2_-based photocatalytic materials have received much attention as a photocatalytic semiconductor. The advantages of TiO_2_-based materials include their biological and chemical inertness, stability, and low cost. On the other hand, pristine TiO_2_ photocatalysts are limited by the fast rate of recombination of photogenerated electron-hole pairs and by their wide band gap, which requires UV irradiation for activation. For these reasons, the modification of TiO_2_ is required [4,5,6].

There are different strategies to prepare modified TiO_2_, involving diverse metal or non-metal doping [7,8,9], multi-phase composition, or combination of TiO_2_ with other semiconductors [10,11]. The presence of surface defects and especially the oxygen vacancies present in TiO_2_ crystalline structure are among the most important factors determining the resulting photocatalytic behavior [12,13]. From this point of view, co-doping by both metal and non-metal atoms is a promising direction of TiO_2_ modification [14]. 

Fluorine doped TiO_2_ photocatalyst can induce visible-light-driven photocatalysis due to the formation of oxygen vacancies. F-TiO_2_-based photocatalysts have been studied in many reactions, for example, in oxidation of 2-propanol [15] and degradation of dyes [16]. The photocatalytic behavior of F-TiO_2_ photocatalysts has been explained by the positive role of fluoride ions to morphology [17], the presence of surface defects forming surface O- species [13], oxygen vacancies [16,18], and higher conductivity [15]. Nevertheless, the role of fluorine on the photocatalytic behaviour of F-TiO_2_-based materials is still under discussion. The problem is partially due to the presence of residual fluoride (after synthesis) adsorbed on the surface of TiO_2_. Surface fluorine has been removed by NaOH which led to the improvement of photocatalytic behavior [13]. 

This article is a natural continuation of our previous research focused on La/TiO_2_ [19] and N-La/TiO_2_ [20] photocatalysts. The role of lanthanum in La/TiO_2_ to its photocatalytic behaviors was reported [19]. Subsequently, the optimal amount of La (ca. 0.2 wt %) was used and the critical parameters of N-La/TiO_2_ playing a role in the decomposition of methanol-water solution [20] were specified. In the current study, a simple method to synthesize TiO_2_ photocatalysts doped with non-metal (fluorine) and metal (lanthanum) elements is used [21]. The physico-chemical properties of F-La/TiO_2_ photocatalysts were characterized by XRD, nitrogen physisorption, UV-Vis, EIS, and XPS. The photocatalytic performance of F-La/TiO_2_ photocatalysts was investigated in the decomposition of methanol-water solution. The attention was focused on the explanation of the contribution of fluorine to the photocatalytic behavior of F-La/TiO_2_ photocatalysts.

## 2. Materials and Methods 

### 2.1. Preparation of Materials

Basic powder materials TiO_2_ and 0.2 wt % La/TiO_2_, were prepared by the sol-gel method within the environment of reverse micelles (Triton X-114 in cyclohexane). Solution containing Triton X-114 (laboratory grade, Sigma Aldrich, Saint Louis, MO, USA), cyclohexane (≥99.5%, Fluka Chemika, Munich, Germany), deionized water with lanthanum (III) nitrate hexahydrate (La(NO_3_)_3_·6H_2_O, 99.99%, Sigma Aldrich, Saint Louis, MO, USA) dissolved in ethanol was stirred for 20 min. Thereafter, titanium (IV) propoxide (98%, Sigma Aldrich, Saint Louis, MO, USA) was added to this solution and mixed again for 20 min. The resulting sols were poured into Petri dishes and left for 24 h at laboratory temperature. The final dry material after gelation was transferred into ceramic crucibles and calcined at 450 °C for 4 h (without heating rate). 

TiO_2_ or La/TiO_2_ powder was added to KBF_4_ (96%, Sigma Aldrich, Saint Louis, MO, USA), which was in 0.05 M 100 mL nitric acid (HNO_3_, 65%, Fluka Chemika, Munich, Germany), the suspension was stirred for 2 h at laboratory temperature and then the mixture was maintained at 180 °C for 24 h. After hydrothermal reaction, the precipitates were filtered, washed by 500 mL deionized water (to get pH = 7) and dried to obtain F-La/TiO_2_ hydrothermal product.

### 2.2. Characterization of Materials

Textural, structural, optical, and electronic properties of all photocatalysts were characterized in detail by nitrogen physisorption [22,23], powder X-ray diffraction (XRD) [22,23], X-ray fluorescence (XRF) [20], diffuse reflectance UV-vis spectroscopy (DRS UV-Vis) [22,24], X-ray photoelectron spectroscopy (XPS) [22], and photoelectrochemical measurements [23] (see Appendix A). Chemical components and the binding energies of CdS/CN were analyzed by XPS (Thermo Scientific ESCALAB 250Xi A1440 system, Waltham, MA, USA). Electrochemical impedance spectroscopy (EIS) analysis was measured by using a CHI 760E electrochemical workstation.

### 2.3. Photocatalytic Test

The photocatalytic decomposition of the methanol-water solution was carried out in a batch photoreactor with a 8 W Hg lamp (λ = 365 nm, see Appendix A) [22]. 

## 3. Results and Discussion

### 3.1. Structural and Optical Properties of La/and F-La/TiO_2_

Table 1 gives the content of La in La/TiO_2_ and F-La/TiO_2_ photocatalysts determined by XRF. The F-La/TiO_2_ photocatalysts contained the real amount of La close to its theoretical amount (0.20 wt % La). La/TiO_2_ contained slightly lower concentration of La in contrast to the theoretical one that could be caused by very low amount of La precursors used in the synthesis. The textural parameters of studied materials are also shown in Table 1. All TiO_2_, La/TiO_2_ and F-La/TiO_2_ photocatalysts possessed similar specific surface area values, close to 80 m^2^ g^−1^.

Figure 1 shows the X-ray patterns of the TiO_2_, La/ and F-La/TiO_2_ photocatalysts. Diffractograms of all photocatalysts showed diffraction lines characteristic of tetragonal modification of anatase (ICDD card no. 00-021-1272; space group I4_1_/amd, lattice constants *a* = *b* = 0.37852 nm and *c* = 0.95139 nm). Coherent domain sizes were in the range of 12.9 and 14.4 nm (Table 2). Any additional diffraction lines reflecting the presence of rutile phase, brookite phase, or lanthanum related phase (pure or oxidic form) were not observed. All materials exhibited approximately the same values of lattice parameters *a* and *c* (Table 2).

The values of the indirect band gap energies of TiO_2_, La/TiO_2_, and F-La/TiO_2_, presented in Table 1, were determined from the dependencies of (α·h·ν)^1/2^ on energy (Figure 2). The band gap energies of the TiO_2_, La/TiO_2_ and F-La/TiO_2_ photocatalysts had the values in the range of 2.85 to 3.01 eV, with TiO_2_ being the lowest band gap energy value. Nevertheless, the mentioned difference is marginal. Thus, the addition of fluorine did not lead to the change of band gap energies.

XPS analysis identified five different elements (Ti, O, C, La, and F) presented on the surface of investigated photocatalysts (Figure 3 and Table 3). The Figure 3a shows the O 1s spectrum of the materials. There are two binding energy peaks at about 529.7 eV and 531.3 eV in all materials, which corresponds to the Ti-O and O–H bonds [25]. With the introduction of F element, a new peak of binding energy appears at about 533.3 eV which corresponds to the H_2_O molecules adsorbed on the surface of the photocatalyst [26], and the intensity of this peak increases with the increase of the amount of F. This phenomenon proves that the doping of F element greatly improves the hydrophilicity of samples [27], which is beneficial for the photocatalytic process. The XPS spectrum of Ti 2p shows two typical binding energy peaks at about 458.6 eV and 464.4 eV due to the Ti 2p_3/2_ and Ti 2p_1/2_ corresponding to +4 valence state of Ti and showing that the elements of Ti firstly existed as Ti^4+^ in the material (Figure 3b) [28]. The doublet peaks derived from spin-orbit splitting of Ti2p affirmed the difference in binding energy is around 5.8 eV. This results also confirm that TiO_2_ phase in the photocatalysts exists only in the presence of anatase, which is the agreement with the XRD analysis. The XPS spectrum in Figure 3c confirms that the La element was successfully doped into La/TiO_2_, 2.8F-La/TiO_2_, and 4.1F-La/TiO_2_ photocatalysts, corresponding to the binding energy peaks of 836.8 and 854.7 eV in the La 3d_5/2_ and La 3d_3/2_, respectively [29]. However, its intensity is marginal and the surface atomic concentration of La is limiting to zero. It is in agreement with our previous works [19], where we also did not observe measurable concentration of La on the surface of La-TiO_2_ materials with low La content. Figure 3d displays the XPS spectrum of F elements of 2.8F-La/TiO_2_ and 4.1F-La/TiO_2_. The binding energy peak appeared at about 684.0 eV and is caused by the F elements adsorbed on the surface of the material (i.e., surface fluorination; T–F bond forming by substitution of OH– group), and the other peak appeared at about 689.0 eV which is caused by F element incorporated into the lattice of TiO_2_ [30]. Fluorine present in the lattice of TiO_2_ results in the formation of Ti^3+^ sites [31]. In order to further analyze the doping of F element in different samples, we conducted integral calculation of the content of F element in these two different forms through Gaussian fitting and the specific results are displayed in Table 4. Due to the fact, that the atom radius of fluorine is resembling oxygen, the substitute of the oxygen atoms joined Ti atoms in the TiO_2_ lattice by atoms of fluorine, which is relatively facile [28]. According to the XPS analysis, the total content of F in the 2.8F-La/TiO_2_ and 4.1F-La/TiO_2_ photocatalysts was determined to be at 2.82% and 4.13%, respectively. 

Figure 4 shows the photocurrent response for the studied photocatalysts. Photocurrent generation can predict behavior of the photocatalyst after irradiation but does not supplement its photocatalytic activity. Since the photocurrent is measured under applied external potential of 1 V, most of the generated electrons are forced to migrate to the ITO foil and measured as current. Simply said, external potential is strongly suppressing the recombination of charge carriers. Based on the results, we can claim 2.8F-La/TiO_2_ photocatalyst generates the highest amount of charge carriers after irradiation with 365 nm. The reason for lower photocurrent values in this study compared to that reported for La-based materials in our works [19,20] is due to the different slits setting. There are two slits, one in front of the monochromator and the second behind the monochromator, which can be set to different values. While previous measurements were done with completely opened slits, the slit behind the monochromator was currently set to 1 mm in order to have more precise wavelength steps. From that reason, the comparison is not possible.

The electrochemical impedance spectroscopy (EIS) technique has been used to research the electron transfer efficiency of the prepared materials (Figure 5). The semicircle in the EIS spectra is caused by contribution from the constant phase element at the photocatalyst/electrolyte interface and the charge transfer resistance. The small semicircle radius of the material indicates a smaller impedance in the photogenerated electron transfer process [32,33,34]. The arc radii of TiO_2_ is shown as the black point plot in Figure 5. And it is clear that the doping of the La element caused the impedance of the sample to be lower than that of the pure TiO_2_. Subsequent increase of conductivity (decrease of semicircle radius) was observed for F-La/TiO_2_ photocatalysts. 2.8F-La/TiO_2_ with the highest conductivity (EIS) also exhibited the highest photocurrent density. After that, the conductivity of photocatalysts decreased with the same order as in the case of the photocurrent response. 

The small fluorine doping concentration (2.8F-La/TiO_2_) caused that the F doped in TiO_2_ favors the transfer of the electrons and represses the recombination of charge in TiO_2_ [35,36,37]. It can be due to the fact that the Ti^3+^ surface states and the oxygen vacancies could capture photoelectrons and transfer them to O_2_ adsorbed on the surface of TiO_2_ [38,39]. In the case of 4.1F-La/TiO_2,_ both the concentration of fluorine and the semicircle were higher. The fluorine doping can act as a center of recombination, which increases the charge recombination in TiO_2_ because the average distance between two trap sites goes down with the growing dopants number [35,38,40].

### 3.2. Photocatalytic Activity of La/and F-La/TiO_2_

The photocatalytic generation of hydrogen from methanol-water solution during UVA illumination in the presence of investigated photocatalysts is shown in Figure 6. The formation of hydrogen is decreasing in order: 2.8F-La/TiO_2_ > 4.1F-La/TiO_2_ ≈ La/TiO_2_ > TiO_2_. The production of hydrogen in the presence of 2.8F-La/TiO_2_ was nearly up to 3 times higher than in the presence of pure TiO_2_. It also should be stressed that the hydrogen yield formed over 2.8F-La/TiO_2_ (Figure 6) is higher than for N-La/TiO_2_ [20].

### 3.3. Properties of F-La/TiO_2_ Playing the Role in Photacatalytic Reaction

In general, the photocatalytic activity of F-La/TiO_2_ photocatalysts can be influenced by several factors, such as (i) band gap energy, (ii) specific surface area, (iii) rate of the recombination of electron-hole pairs, (iv) presence of oxygen vacancies, (v) amount of lattice and surface O species, and (vi) phase composition. In our case, F-La/TiO_2_ photocatalysts and pure TiO_2_ exhibited approximately the same structure, specific surface area (Table 1), and band gap energy (Table 1 and Figure 2).

The hydrogen yield (Figure 6) clearly increased with increasing photocurrent response (Figure 4) and increasing conductivity (Figure 5). The increasing value of both photocurrent and conductivity can reflect a decrease of the recombination probability of photogenerated electron-hole pairs. Figure 7 shows the correlation between the amount of hydrogen and the photocurrent response. The trend of photocatalytic activity of F-La/TiO_2_ corresponds to their photocurrent responses (Figure 7b). Low photocurrent means lower amount of generated electron-hole pair after irradiation. The 2.8F-La/TiO_2_ photocatalyst showed both the highest photocatalytic activity and the photocurrent response as well. This is in agreement with many other authors [19,41,42,43].

The value of photocurrent response and conductivity can reflect the combination of two factors, i.e., the amount of fluorine substituted into the oxygen sites of the TiO_2_ lattice and the amount of surface O-sites.

Combining with the XPS results (Figure 3d), it was affirmed that atoms of fluorine was substituted also into the oxygen sites of the TiO_2_ lattice. The charge imbalance is formed when -1 F ions replaced the -2 oxygen sites. The excess positive charge is possible to be neutralized by creating hydroxide ions which form from surface adsorbed hydroxyl groups. Consequence to this, more reactive oxygen species as hydroxyl radicals and superoxide free radicals can be created, which show the existence of oxygen vacancies. These oxygen species have strong oxidative capability and play a significant role in the photocatalytic degradation of methanol-water solution [18,44]. In agreement with that, 2.8F-La/TiO_2_ possessed a higher ratio of lattice to surface fluorine species (Table 4) and higher hydrogen yield (Figure 7), as well as the highest photocurrent density and conductivity (EIS).

Previously, we observed for La-N/TiO_2_-based photocatalysts that the hydrogen yield increased with increasing amount of oxygen vacancies and decreasing amount of surface oxygen species (surface lattice O species and hydroxyl groups, XPS) [20]. In principle, the amount of produced hydrogen yield (Figure 7) increased with decreasing content of surface O-sites also for La-F/TiO_2_ (Table 3), but the clear correlation could not be done due to the different population of surface C-species among the individual materials. Thus, we calculated O/Ti surface atomic ratio (Table 3), with the lowest value for 2.8F-La/TiO_2_. The O/Ti surface atomic ratio was lower for both F-La/TiO_2_ photocatalysts than for TiO_2_ and La/TiO_2_ which show to decrease the amount of surface oxygen species after F-loading.

## 4. Conclusions

Based on the above characterizations and photocatalytic activity, co-doping by lanthanum and fluorine proved to be an effective method to prepare the highly photoactive TiO_2_-based photocatalysts for production of hydrogen from photocatalytic decomposition of methanol-water solution. 

The anatase TiO_2_ photocatalysts with different F amounts and the similar La amount (0.13–0.21 wt %) were prepared by sol-gel method and in detail characterized by various methods. The highest hydrogen yield during the photocatalytic decomposition of methanol-water solution exhibited the 2.8 F-La/TiO_2_ photocatalyst.

The prepared F-La/TiO_2_ (3.36 and 4.45 at % fluorine content) photocatalyst showed enhanced photocatalytic activity in decomposition of methanol-water solution under 365 nm illumination, the hydrogen production in the presence of 2.8F-La/TiO_2_ and 4.1F-La/TiO_2_ photocatalysts was nearly up to 3 and 2 times higher in comparison to the pure TiO_2_, respectively. The photocatalytic performance of F-La/TiO_2_ increased with increasing photocurrent response and conductivity. These results indicated that the fluorine ions were successfully incorporated into the lattice of TiO_2_. Moreover, the luorine and lanthanum co-doping of TiO_2_ photocatalyst can effectively decrease the recombination probability of photogenerated electron-hole pairs.

## Figures and Tables

**Figure 1 materials-12-02867-f001:**
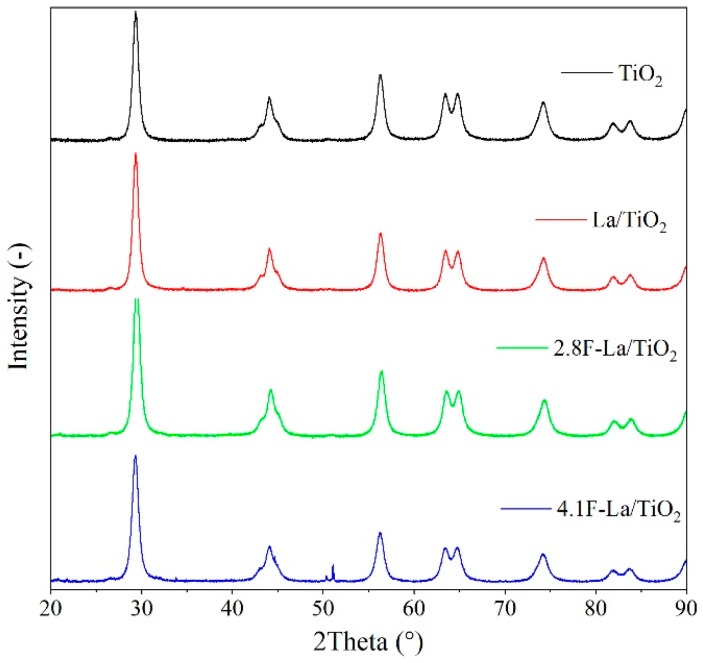
XRD patterns of investigated photocatalysts.

**Figure 2 materials-12-02867-f002:**
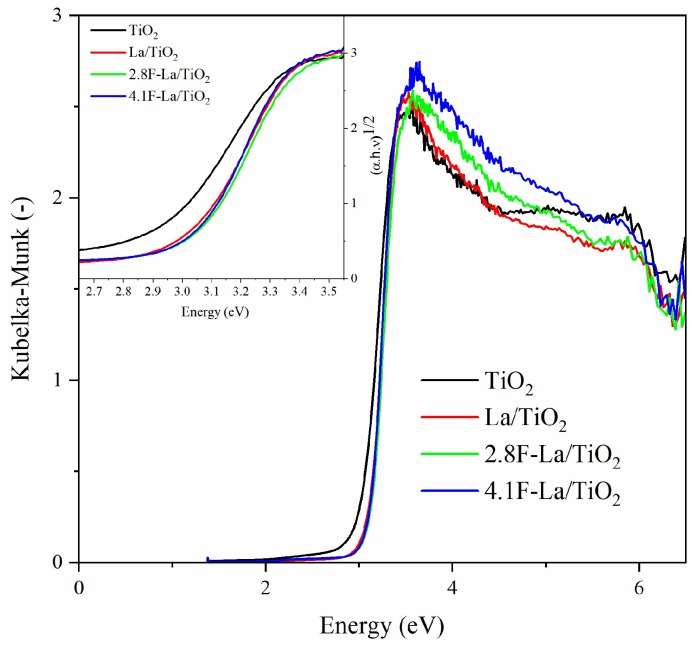
UV-Vis DRS spectra of investigated photocatalysts. The cut-out contains the Tauc plot and the determination of the indirect band gap energy values.

**Figure 3 materials-12-02867-f003:**
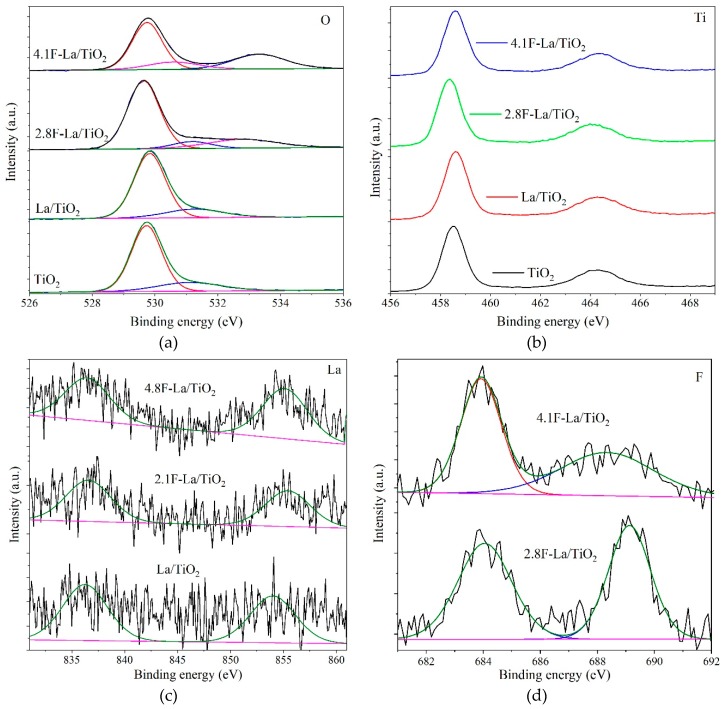
XPS spectra for oxygen (**a**), titanium (**b**), lanthanum (**c**) and fluorine (**d**) of investigated photocatalysts.

**Figure 4 materials-12-02867-f004:**
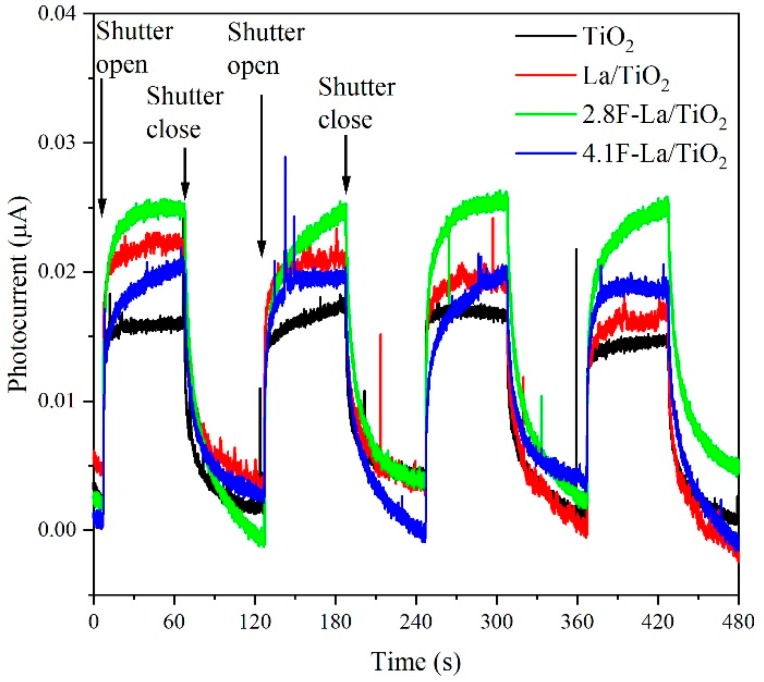
Photocurrent responses of prepared photocatalysts under applied external potential 1 V and irradiated under 365 nm.

**Figure 5 materials-12-02867-f005:**
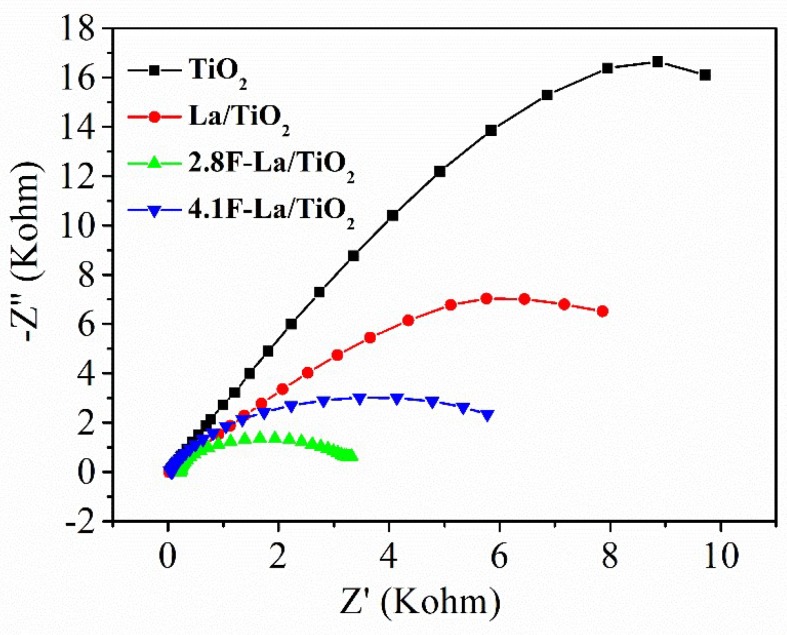
Electron transfer efficiency of investigated photocatalysts.

**Figure 6 materials-12-02867-f006:**
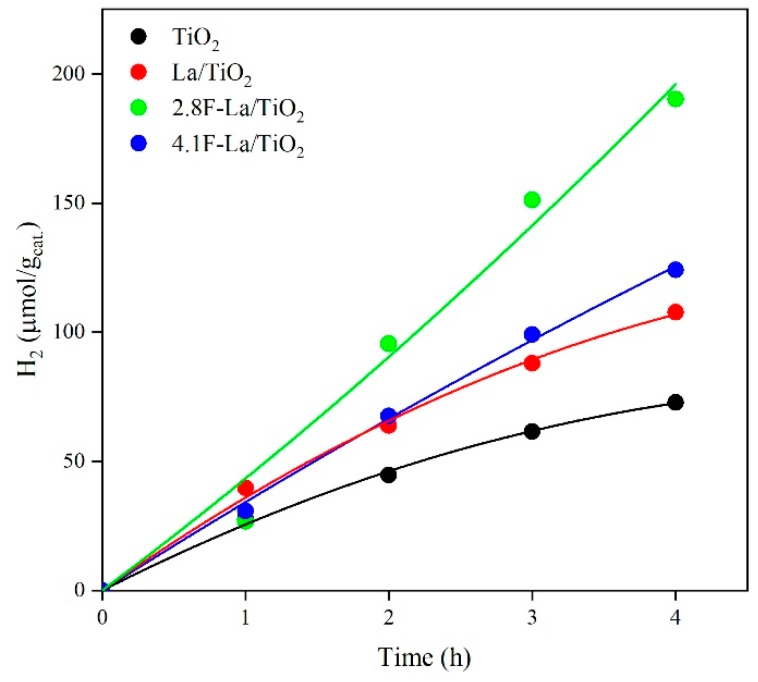
Generation of hydrogen in the photocatalytic degradation of methanol-water solution in the presence of the investigated photocatalysts.

**Figure 7 materials-12-02867-f007:**
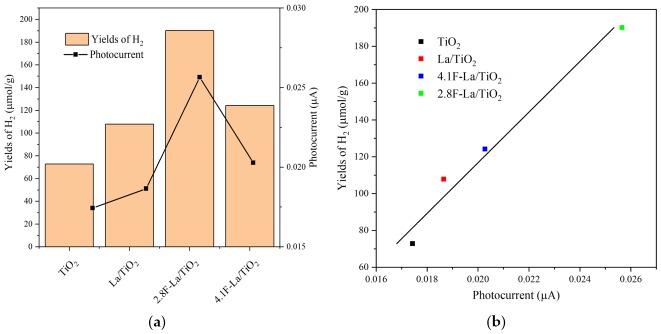
Correlation between the photocatalytic activity in the photocatalytic decomposition of methanol-water solution (**a**) and the hydrogen yields dependence on photocurrent (**b**) in the presence of the investigated photocatalysts and current generation. Current responses were obtained at 365 nm under external potential of 1.0 V.

**Table 1 materials-12-02867-t001:** Composition, textural, and optical properties of investigated photocatalysts.

Photocatalyst	XRF	Textural Properties	DRS UV-vis
	The Content of La (wt %)	S*_BET_* (m^2^.g^−1^)	Indirect Band Gap (eV)
TiO_2_	-	80	2.85
La/TiO_2_	0.13	73	2.95
2.8F-La/TiO_2_	0.20	84	3.01
4.1F-La/TiO_2_	0.22	79	3.00

**Table 2 materials-12-02867-t002:** Structural and microstructural properties of investigated photocatalysts.

Photocatalyst	Anatase Crystallite Size nm	Lattice Parameters	
		*a* nm	*c* nm
TiO_2_	14.4	0.3788	0.9510
La/TiO_2_	14.7	0.3787	0.9504
2.8F-La/TiO_2_	13.3	0.3787	0.9506
4.1F-La/TiO_2_	12.9	0.3788	0.9505

**Table 3 materials-12-02867-t003:** Surface concentration of Ti, O, F, and C elements determined by XPS.

Photocatalyst	Ti (at %)	O (at %)	F (at %)	C (at %)	O/Ti
TiO_2_	31.27	60.44	0	8.30	1.93
La/TiO_2_	30.84	59.58	0	9.58	1.93
2.8F-La/TiO_2_	26.26	49.46	2.83	21.45	1.88
4.1F-La/TiO_2_	30.63	58.31	4.14	6.93	1.90

**Table 4 materials-12-02867-t004:** Surface concentration of surface fluorine and lattice fluorine determined by XPS.

Photocatalyst	Content of F (at %)	Surface Fluorine (at %)	Lattice Fluorine (at %)	Ratio of Lattice and Surface Fluorine
2.8F-La/TiO_2_	2.83	1.47	1.36	0.931
4.1F-La/TiO_2_	4.14	2.22	1.92	0.865

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
