# Peer review of "The Role of Fluorine in F-La/TiO2 Photocatalysts on Photocatalytic Decomposition of Methanol-Water Solution"

_materials, 2019, doi:10.3390/ma12182867_

Round 1

Reviewer 1 Report

The paper presents clearly the research steps and results. It may be published in the present form A

A small editing error that I noticed : at line 132 2.8Fla/ TiO2 should be 2.8F La/TiO2

Author Response

The paper presents clearly the research steps and results. It may be published in the present form A.

A small editing error that I noticed: at line 132 2.8Fla/ TiO2 should be 2.8F La/TiO2

Comments:

We thank reviewer for this note. This error was corrected.

Reviewer 2 Report

Reviewer’s Report

The role of fluorine in F-La/TiO2 photocatalysts on photocatalytic decomposition of methanol-water solution

Manuscript ID: materials-570070

In this manuscript, the authors investigated the F-La/TiO2 catalysts for the photocatalytic decomposition water-methanol solution. They reported that the prepared F-La/TiO2 showed enhanced photocatalytic activity in decomposition of methanol-water solution in comparison with the pure TiO2. The results presented in the manuscript could be publishable after the following minor revisions.

Provide the EDS color mappings to know the distribution of elements in the catalysts. Page 4, line 119, correct Figure 4 as Figure 3. Provide a, b, c, d for Figure 3. There are several typos that need to be corrected before publication. The following papers can be cited in the manuscript.

        Catalysis Communications 113 (2018) 1-5; Journal of Industrial and            Engineering Chemistry 20 (2014) 444-453

Author Response

Provide the EDS color mappings to know the distribution of elements in the catalysts.

Comment:

Unfortunately, we do not have a possibility to measure EDS color mapping.

Page 4, line 119, correct Figure 4 as Figure 3. 

Comment:

This error was corrected.

Provide a, b, c, d for Figure 3. There are several typos that need to be corrected before publication. 

Comment:

The a, b, c, d for Figure 3 were added.

The following papers can be cited in the manuscript.

Catalysis Communications 113 (2018) 1-5; Journal of Industrial and Engineering Chemistry 20 (2014) 444-453

Comment:

The both publication were added to the manuscript.

Reviewer 3 Report

The manuscript includes an interesting concern related to the effect of La-F doping on TiO2 over the effectiveness of this oxide in the hydrogen production from a methanol- water solution. In my opinion, this work presents interesting results, the manuscript has a good writing and the results obtained in photocatalytic activity tests are suitably supported by characterization outcomes. I recommend the publication of the manuscript after minor revisions, authors are suggested to attend to the next recommendations:

A complete description about the batch reactor and reaction conditions must be reported in section 2.3.

What 2.8 and 4.1 are meaning in Table 1?

What is the reason why the band gap value of TiO2 increases after F and La addition?

The band gap value of TiO2 in Anatase form has been reported to be over 3,0eV, why the value of this material is lower than the usual reported value?

Could the authors explain in deep the next sentence, please? Why authors said that “…increasing the hydrophilicity of samples is beneficial for the photocatalytic process”.

Some discussion about the O/Ti ratio values (Table 3), should be included in the revised manuscript.

Quality of Figure 7 must be improved.

In experimental section, authors have indicated that the photocatalysts were prepared by sol-gel method, but in the second conclusion, they have indicated that hydrothermal method was employed to obtain the doped materials. Can the authors clarify it, please?

Author Response

A complete description about the batch reactor and reaction conditions must be reported in section 2.3.

Comment:

The complete description about the batch reactor and reaction conditions are reported in the Supplementary materials. We decided for this way in order to we avoided the similarity with our previous work.   

What 2.8 and 4.1 are meaning in Table 1?

Comment:

The 2.8 and 4.1 mean the content of fluorine in at.%. It is describe in the Table 3.

What is the reason why the band gap value of TiO2 increases after F and La addition?

Comment:

We specify it in the text. There is marginal change of band gap energies for TiO2, La/TiO2 and F-La/TiO2. Thus, the addition of Fluorine did not led to the change of band gap energies.

The band gap value of TiO2 in Anatase form has been reported to be over 3,0eV, why the value of this material is lower than the usual reported value?

Comment:

We agree that the band gap energy of Anatase is 3.2 eV, so this value is slightly higher in contrast to that reported in this manuscript (3.0 eV). The difference could be explained by different values of crystallite size of TiO2.

Could the authors explain in deep the next sentence, please? Why authors said that “…increasing the hydrophilicity of samples is beneficial for the photocatalytic process”.

Comment:

The photoinduced hydrophilicity and photocatalytic H2 evolution from water are closely related to the adsorption of water, a polar molecule [27].

[27] Tang, J.; Quan, H.; Ye, J. Photocatalytic Properties and Photoinduced Hydrophilicity of Surface-Fluorinated TiO2. Chem. Mater. 2007, 19, 116-122, doi:10.1021/cm061855z.

Some discussion about the O/Ti ratio values (Table 3), should be included in the revised manuscript.

Comment:

The O/Ti surface atomic ratio was lower for both F-La/TiO2 photocatalysts than for TiO2 and La/TiO2 that shows to the decrease of the amount of surface oxygen species after F-loading. 

Quality of Figure 7 must be improved.

Comment:

The quality of Figure 7 was improved.

In experimental section, authors have indicated that the photocatalysts were prepared by sol-gel method, but in the second conclusion, they have indicated that hydrothermal method was employed to obtain the doped materials. Can the authors clarify it, please?

Comment:

We thank the reviewer for this note. This mistake in conclusion was corrected.
